# Topology and Dynamics of Transcriptome (Dys)Regulation

**DOI:** 10.3390/ijms25094971

**Published:** 2024-05-02

**Authors:** Michel Planat, David Chester

**Affiliations:** 1Institut FEMTO-ST CNRS UMR 6174, Université de Franche-Comté, 15 B Avenue des Montboucons, F-25044 Besançon, France; 2Quantum Gravity Research, Los Angeles, CA 90290, USA; davidc@quantumgravityresearch.org

**Keywords:** group theory, character variety, Painlevé equations, transcriptome, microRNAs, diseases, cancer research

## Abstract

RNA transcripts play a crucial role as witnesses of gene expression health. Identifying disruptive short sequences in RNA transcription and regulation is essential for potentially treating diseases. Let us delve into the mathematical intricacies of these sequences. We have previously devised a mathematical approach for defining a “healthy” sequence. This sequence is characterized by having at most four distinct nucleotides (denoted as nt≤4). It serves as the generator of a group denoted as $f_{p}$. The desired properties of this sequence are as follows: $f_{p}$ should be close to a free group of rank nt−1, it must be aperiodic, and $f_{p}$ should not have isolated singularities within its $SL_{2}\left(C\right)$ character variety (specifically within the corresponding Groebner basis). Now, let us explore the concept of singularities. There are cubic surfaces associated with the character variety of a four-punctured sphere denoted as $S_{2}^{4}$. When we encounter these singularities, we find ourselves dealing with some algebraic solutions of a dynamical second-order differential (and transcendental) equation known as the Painlevé VI Equation. In certain cases, $S_{2}^{4}$ degenerates, in the sense that two punctures collapse, resulting in a “wild” dynamics governed by the Painlevé equations of an index lower than VI. In our paper, we provide examples of these fascinating mathematical structures within the context of miRNAs. Specifically, we find a clear relationship between decorated character varieties of Painlevé equations and the character variety calculated from the seed of oncomirs. These findings should find many applications including cancer research and the investigation of neurodegenative diseases.

## 1. Introduction

Self-regulation of an animal organism, also-called homeostasis, depends on several factors. At a macroscopic scale, pH, temperature, the amount of oxygen and carbon dioxide in the blood and so on in living organisms self-adjust to fight against threats present in the environment and keep a stable physiology.

In mammals, the main organs involved in homeostasis are the hypothalamus that controls body temperature and circadian cycles; the pituitary gland that rates the secreting hormones, the lungs that regulate the balance between blood oxygen and the carbon dioxide; the skin that protects against outer pathogens; the muscles in the circulatory, gastrointestinal and blood systems; the kidneys that regulate water and salts in the blood and the amount of urine due to hormones; the liver which acts as a factory for bile, glucose, hemoglobin, ammonia and toxins; the pancreas that produces the hormones and enzymes for digestion; and finally the brain that controls behavior whether conscious or unconscious. Such macroscopic features of homeostasis are well summarized by Claude Bernard in their 19th-century lectures [1].

A genome-scale homeostasis exists as well. Gene expression (GE) is the process by which information from a gene is used for the synthesis of end products such as the production of proteins or non-coding ribonucleic acid (RNA). GE is the most fundamental level at which the genotype gives rise to observable traits called the phenotype. At all steps, GE is regulated by transcription, translation and post-transcriptional changes in the production of proteins. Such microscopic homeostasis for creating messenger mRNA is a complex machinery that makes use of non-coding genes transcribed as precursors that undergo further processing [2].

Rules for genome-scale homeostasis similar to the constancy rules of macroscopic homeostasis are not yet firmly established. We mention previous work in this direction [3,4,5]. In our previous publications, we started to unveil some of these rules by using a group theoretical approach. Since the genome is digital with its four nucleotides A, U/T, G and C as letters, there are important short sequences as the consensus sequence of a transcription factor, a TATA box, a polyadenymation signal or a microRNA (miRNA) whose structure should obey group theory and associated character variety rules. Otherwise, a disease is in sight [6,7,8]. Chemical modifications in the environment also controls the metabolism of GE. For instance, RNA methylation with $N^{6}$-methymadenosine ($m^{6}A$) is documented in our recent paper [8].

For studying the aforementioned key sequences, we employ infinite (finitely generated) groups denoted by $f_{p}$, and their representations over the matrix group $SL_{2}\left(C\right)$, where the entries of matrices are complex numbers [6]. The importance of this group extends across all fields of physics as it represents a space-time-spin group.

Our crucial observation is that an $f_{p}$ group associated with a “healthy” sequence usually approximates a free group $F_{r}$, where the rank *r* equals the number of distinct nucleotides minus one. A sequence deviating from this may suggest a potential dysregulation leading to a disease. However, an $f_{p}$ group closely resembling a free group does not provide sufficient assurance against a disease. Additional examination of the $SL_{2}\left(C\right)$ representations of $f_{p}$ termed the character variety, specifically its Groebner basis G, is necessary. The Groebner basis comprises a set of surfaces. A surface within G containing isolated singularities indicates a potential disease that can be identified specifically, e.g., relating to an oncogene or a neurological disorder ([6], Figure 6 and Tables 2–4).

An additional attribute of “healthy” sequences, which leads to a group $f_{p}$ approximating the free group $F_{r}$ and not mentioned in [6], is their connection to aperiodicity. Schrödinger’s book [9] proposes aperiodicity of living “crystals”. Our papers [7,8] characterize some aperiodic DNA/RNA sequences. However, in this paper, we do not focus on this aspect of genome scale homeostasis.

Our new goal in the current paper is to feature the dynamical equations sustaining some of the algebraic surfaces contained in the Groebner basis attached to the group $f_{p}$ of the considered short sequence. This task was started in [10] using the list of algebraic solutions of the Painlevé VI (PVI) Equation [11]. We made use of the PVI equation and the cubic algebraic surfaces contained in its character variety in the context of miRNAs. We continue along these lines by looking at cubic surfaces contained in the character variety of Painlevé equations of lower index such as PV and PIV. As explained below, this is a crucial step since the topology of the character variety of PV and PIV is wilder than that of PVI, and the singularities of the corresponding surfaces are more irregular than those with PVI [12,13]. The occurrence of surfaces related to PV and PIV may signify a more severe disease than a surface related to PVI.

In Section 3, we investigate the topology of Painlevé equations and their related character variety. We also remind readers how the character variety of a group $f_{p}$ may contain simply singular surfaces of the Painlevé type. In Section 2, we apply these techniques to miRNAs recognized as oncomirs or tumor suppressors. Our results are in accordance with our aforementioned view of health versus disease. Section 4 summarizes our approach.

## 2. Results and Discussion

The theory developed in the previous section is applied to sequences determined by the seed of an miRNA. Animal miRNAs are able to recognize their target mRNAs by using as few as six to eight nucleotides (the seed region) at the $5^{\prime }$ end of the miRNA (for a -5p strand) or at the $3^{\prime }$ end of the miRNA (for a -3p strand). A given miRNA may have hundreds of different mRNA targets, and a given target might be regulated by multiple miRNAs. Thus, it is not surprising that a simple correlation between an miRNA and a disease cannot be found. In the following, the selected miRNAs occur in the context of cancer research [14]. We mention that there are repositories where useful links between diseases and non-coding RNAs such as miRNAs are listed ([15,16]. The seeds used for our calculations of the group $f_{p}$ are taken from the on-line microRNA repository [17,18].

Our results are summarized in Table 1 for miRNAs generally considered as oncomirs and Table 2 for miRNAs generally considered as tumor suppressors. Of course the boundary between the two types is not sharp and depends on the type of disease under examination. We remind readers from the introduction that dysregulation may occur when either the $f_{p}$ group generated by the seed sequence is away from the free group $F_{1}$ (for a seed with two distinct nucleotides), away from the free group $F_{2}$ (for a seed with three distinct nucleotides) or away from the free group $F_{3}$ (for a seed with three distinct nucleotides). The number of conjugacy classes of subgroups of a given index for the group $f_{p}$ is abbreviated as ‘card seq’. The first step is to check if the card seq is close to that of a free group $F_{r}$, r=1, 2 or 3 or not. In the later case, the sequence may predict a risk of disease.

The second step is to check the structure of the character variety associated with $f_{p}$, more precisely the structure of its Groebner basis G. We focus on cubic surfaces belonging to G and, if they contain an isolated singularity, we determine the A-D-E type (abbreviated as ADE type). From the introduction, we claim that dysregulation may occur if such an isolated singularity exists in a cubic surface in G. Finally, we can check if the simply singular surface belongs to the Painlevé class (PVI, PV or PIV), e.g., in Table 1, the notation $f^{(4A_{1}})$: PVI means that the Cayley cubic of ADE type $4A_{1}$ is a solution of equation $xyz+x^{2}+y^{2}+z^{2}-4=0$ associated with the character variety of PVI whose topology is the four-punctured sphere (as in Figure 1). In Table 1, the notation $f^{\left(A_{2}\right)}$: PV for the surface $xyz+x^{2}+y^{2}+1$ means that the ADE type is $A_{2}$ and that the surface is associated with the PV equation.

All oncomirs in Table 1 have an $f_{p}$ such that the corresponding G contains simply singular cubic surfaces whatever the card seq of $f_{p}$ is that of a free group or not. Thus, a correlation between our approach and the risk of the disease is well established.

In Table 2, only five miRNAs generally considered as tumor suppressors have an $f_{p}$ either away from the free group $F_{2}$ or with a G containing simply singular cubic surfaces. The remaining miRNAs have an $f_{p}$ like the free group $F_{2}$ and show no simply singular surface in G. It is well known that a microRNA may act as an oncogen for a given type of cancer or as a tumor suppressor for another type, e.g., let-7a-5p is a regulator for tumors in many cancers but may also unfavorable in lung cancer.

We do not investigate miRNA seeds comprising four distinct nucleotides. It is straightforward to check if the corresponding $f_{p}$ group is close to the free group $F_{3}$ or not and if the seed sequence is aperiodic or not. However, the Groebner basis is more difficult to obtain, and its structure does not allow us to differentiate between ‘healthy’ and ‘disruptive’. Further progress is necessary ([10], Section 5.1).

Details and references for each considered miRNA are given in the subsections below. Most cubics appearing in the Groebner basis attached to the $f_{p}$ group of the seed are degree 3 Del Pezzo surfaces (denoted as DP3).

### 2.1. Human Micrornas That Mainly Act as Oncomirs

In this subsection, we provide details about the results summarized in Table 1. The first five miRNAs have two disinct nucleotides in their seed. Their card seq may be that of the free group $F_{1}$ of rank one or correspond to a group away from $F_{1}$. All of them have a Groebner basis containing DP3 simply singular cubic surfaces. The other selected oncomirs have three distinct nucleotides in their seed. Their card seq may be that of the free group $F_{2}$ of rank one or correspond to a group away from $F_{2}$. All of them have a Groebner basis containing DP3 simply singular cubic surfaces. Some of the surfaces correspond to a Painlevé equation PVI (${\mathrm{PVI}}_{a}$ means that we are dealing with an algebraic solution of PVI), and some surfaces correspond to PV equation. These results confirm our approach that a disease is in sight when either the card seq is away from a free group or the Groebner basis contains surfaces with isolated singularities. As already mentioned, PV is a wilder singular form than PV, and we can expect a more severe disease in this case.

#### 2.1.1. The Oncomir miR-21-3p

The clinical importance of miR-31-3p in human cancers was investigated in [20]. miR-21 expression is different among cancers, and the expression level of miR-21 is primarily increased to promote tumor progression. Upregulation of miR-21 results in drug resistance and radio-resistance in various cancers.

The seed aacacca of miR-21-3p contains two distinct nucleotides a and c. The associated card seq is the sequence [1,1,1,2,1,3,3,1,2,2,⋯] that is away from that of the free group $F_{1}$. According to our approach, this feature may be associated with a disease. The Groebner basis associated with the character variety of the seed is simple enough to merit an explicit display as
$$\begin{array}{cc} & G_{miR-21-3p}=\left(yz^{2}-x^{2}-z^{2}-2y+4\right)\left(x^{3}-y^{2}z+xy-3x+z\right)\\ & \left(z^{3}-x+3z\right)\left(xz-z^{2}-y+2\right). \end{array}$$
The two cubic surfaces defining $G_{miR-21-3p}$ are degree 3 Del Pezzo (DP3) of the ADE type $f^{\left(A_{3}\right)}$ and $f^{\left(A_{2}\right)}$ (shown in Figure 2: left). The third factor is a curve, and the last one is a quadric not playing a role in our singularity analysis.

#### 2.1.2. The Oncomir miR-204-5p

Abnormal expression of miR-204-5p plays a contrasting role in cancer [21,22]. A decrease expression often due to hypermethylation favors the progression of tumors, while an increase expression may also be associated with prostate, breast or ovarian cancers. However, miR-204-5p also has a role in a variety of tumors by regulating metastasis and apoptosis.

The seed ucccuuu of miR-204-5p contains two distinct nucleotides u and c. The associated card seq is the sequence [1,1,2,5,5,25,43,123,⋯] that is away from that of the free group $F_{1}$. According to our approach, this feature may be associated with a disease. The Groebner basis associated with the character variety of the seed is simple
$$\begin{array}{cc} & G_{miR-204-5p}=\left(x-1\right)\left(x+1\right)\left(xz^{2}-y^{2}-z^{2}-2x+4\right)\\ & \left(y^{3}-x^{2}z+xy-3y+z\right)\left(xyz-x^{2}-y^{2}-yz+x+2\right). \end{array}$$
The three cubic surfaces defining $G_{miR-204-5p}$ are DP3 of the type $f^{\left(A_{3}\right)}$, $f^{\left(A_{2}\right)}$ and $f^{\left(A_{2}\right)}$ (shown in Figure 2), respectively.

#### 2.1.3. The Oncomir miR-126b-5p

MicroRNA miR-126b-5p promotes tumor cell proliferation, metastasis and invasion by targeting the enzyme TD02 in hepatocellular carcimona, one of the most common gastrointestinal malignancies, with the third highest mortality rate [23].

The seed auuauua of miR-204-5p contains two distinct nucleotides a and u. The associated card seq is the sequence [1,2,3,2,8,7,10,18,28,27,88,⋯] which corresponds to the modular group, denoted as $H_{3}$. According to our approach, this feature may be associated with a disease. The Groebner basis associated with the character variety of the seed
$$\begin{array}{cc} & G_{miR-126b-5p}=\left(y^{3}-x^{2}z+xy-3y+z\right)\left(y^{2}z-xy-xz+y-z\right)\\ & \left(xyz-x^{2}-y^{2}-yz+x+2\right)\left(xz^{2}-y^{2}+z^{2}-2yz\right)\left(yz^{2}-xz-y-z\right). \end{array}$$

The first three cubic surfaces defining $G_{miR-126b-5p}$ are DP3 of the type $f^{\left(A_{2}\right)}$ (an isolated singularity of the ADE type $A_{2}$) or $f^{\left(A_{1}A_{2}\right)}$ (an isolated singularity of the ADE type $A_{1}A_{2}$). The last two cubic surfaces are rational scrolls devoid of an isolated singularity.

#### 2.1.4. The Oncomir miR-1908

miR-1908-5p is aberrantly expressed in many diseases, especially cancer, being widely involved in a variety of cellular behaviors, including cell proliferation, cell differentiation, apoptosis, cancer cell invasion and metastasis, and extracellular vesicle secretion [24]. miR-1908-3p promotes the breast cancer cells proliferation and metastasis by suppressing several genes [25]. miR-1908 is also involved in the pathoetiology of bipolar disorder, myocardial infarction, obesity, renal fibrosis, rheumatoid arthritis and scar formation [26].

The seed ggcgggg of miR-1908-5p contains two distinct nucleotides c and g. The associated card seq is that of the free group $F_{1}$ with a single generator. The attached Groebner basis contains the Cayley cubic $f^{\left(4A_{1}\right)}$: PVI.

The seed cggccgc of miR-1908-3p contains two distinct nucleotides c and g. The associated card seq is that of the group $H_{3}$ already encountered for miR-126-5p. The attached Groebner basis contains DP3 surfaces of ADE type $A_{2}$ and $A_{1}A_{2}$.

#### 2.1.5. The Oncomir miR-155-3p

miR-155 is a key miRNA in both immunity and cancer, and its miRNA strand, miR-155-3p, has been functionally implicated in both areas despite its low abundance (about 2%) compared to the -5p strand. An increased miR-155-3p strand percentage in cancerous tissues may indicate a specific dysregulation of miR-155 [27].

The seed uccuaca of miR-155-3p contains three distinct nucleotides a, c and u. The associated card seq is the sequence [3,10,51,164,1230,7829,59,835,491,145⋯] that is away from that of the free group $F_{2}$ of rank 2. According to our approach, this feature may be associated with a dysregulation and a disease.

The character variety is much more complicated than in the case of seeds with two distinct nucleotides. However, it is not difficult to identify that the surface $xyz+y^{2}+z^{2}-d=0$, with d=4 or 6, of ADE type $A_{2}$, is part of the game.

Additionally, the Groebner basis contains the surface $2xyz-2xz-2x+y^{2}-z^{2}-2z-1=0$, whose ADE type is $A_{1}A_{2}$.

#### 2.1.6. The Oncomir miR-9-5p

miR-9-5p is a well-known promotor or suppressor of cancer metastasis. miR-9 mainly acts as a promoter of metastasis capability in breast cancer, osteosarcoma, prostate cancer and bladder cancer, while it plays the opposite roles in other cancers such as colorectal, nasopharyngal carcimona, melanoma and gastric and brain cancers ([28], Table 1).

The seed cuuuggu of miR-9-5p contains three distinct nucleotides c, u and g. The associated card seq is that of the free group $F_{2}$.

The Groebner basis for the character variety attached to the seed contains the cubic surface $xyz+x^{2}+z^{2}+10=0$ which is of the type $f^{\left(A_{2}\right)}$: ${\mathrm{PV}}^{deg}$. The Painlevé type is obtained with $\omega_{1}=\omega_{2}=0$ and the change of variables $x\to \sqrt{10}x$, $z\to \sqrt{10}z$. The Groebner basis also contains the DP3 cubic surface, $xyz+x^{2}+y^{2}+z^{2}-2y+z+1=0$ (see Figure 3: Left), and similar surfaces related to the solutions of the PVI equation. Such surfaces are non-algebraic that is not in the list provided in ([11], Table 4).

#### 2.1.7. The Oncomir miR-146b-3p

It is known that miR-146b-3p is one important regulator of the proliferation of pancreatic cancer stem cells [29].

The seed cccugug of miR-146b-3p contains three distinct nucleotides c, g and u. The associated card seq is the sequence [3,13,92,499,5955,77,873⋯] that is away from that of the free group $F_{2}$. This feature may be associated with a disregulation and a disease.

The Groebner basis attached to the seed contains DP3 cubic surfaces such as the surface $xyz+y^{2}+z^{2}-2=0$ of ADE type $A_{2}$ and the surface $xyz+y^{2}+z^{2}-2y-2z+1=0$ the ZDE type $A_{2}A_{1}A_{1}$ and Painlevé form PV, shown in Figure 3: right.

#### 2.1.8. The Oncomir miR-19a-5p

miR-19a-5p is part of the miR-17-92 cluster: a NF-kB regulated family of miRNAs that promotes cell proliferation, oncogenic transformation and evasion of apoptosis. High levels of miR-19a-5p correlate with poor prognosis in pancreatic cancer [30].

The seed guuugc of miR-19a-5p contains three distinct nucleotides c, g and u. The associated card seq is that of the free group $F_{2}$.

The Groebner basis attached to the seed contains the DP3 cubic surface $xyz+x^{2}+y^{2}+z^{2}-6$ which corresponds to a solution of the PVI equation devoid of an isolated singularity. This solution is the (octahedral) algebraic solution 30 in [11]. A parametric plot for the modulus of this solution is in [10], Figure 5.

Other DP3 cubic surfaces in the Groebner basis are $xyz+y^{2}-8=0$ of type $f^{\left(A_{2}A_{2}\right)}$ and $xyz+y^{2}-xy-yz-2xz+z-1=0$ of type $f^{\left(A_{1}A_{1}\right)}$.

#### 2.1.9. The Oncomir miR-181a-5p

The dysregulation of miR-181a-5p has been implicated in various types of cancer and functions as an oncomir or tumor inhibitor [31].

The seed acaucaa of miR-181a-5p contains three distinct nucleotides a, c and u. The associated card seq is that of the free group $F_{2}$. The Groebner basis attached to the seed contains DP3 cubic surfaces such as the surface $xyz+x^{2}+z^{2}-2=0$, $xyz+x^{2}+z^{2}-4=0$ and $xyz+x^{2}+z^{2}-x-2z-2=0$ which are all the ADE type $A_{2}$. It also contains the surface $xyz+x^{2}+z^{2}-x=0$ of ADE type $A_{4}$.

#### 2.1.10. The Oncomir miR-15-5p

miR-15-5p participates in the pathogenesis of several cancers as well as non-malignant conditions, such as abdominal aortic aneurysm, Alzheimer’s and Parkinson’s diseases. Dysregulation of miR-15b-5p in clinical samples has been associated with poor outcome in different kinds of cancers [32].

The seed agcagca of miR-15-5p contains three distinct nucleotides a, c and g. The associated card seq is that of the free group $F_{2}$. The Groebner basis attached to the seed contains a DP3 cubic surface $xyz+x^{2}+y^{2}-4y+1=0$ which is of type $f^{\left(A_{2}\right)}$: PV as well of other surfaces of the PV type.

#### 2.1.11. The Oncomir miR-569

miR-569 expression levels are markedly downregulated in lung cancer cells. miR-569 also contributes to ovarian and breast cancer cell survival and proliferation [33].

The seed guuaaug of miR-569 contains three distinct nucleotides a, g and u. The associated card seq is the sequence [3,10,51,164,1230,7829⋯] that is away from that of the free group $F_{2}$. This feature may be associated with a disregulation and a disease. The Groebner basis attached to the seed contains the DP3 cubic surfaces $xyz+x^{2}+y^{2}-4=0$ which is of the ADE type $A_{2}$ and $y^{2}z+2xy-2z^{2}-2x+z=0$ which is of type $A_{4}$.

#### 2.1.12. The Oncomir miR-133b

miR-133b originally defined as a canonical muscle-specific microRNA. It has been found to show abnormal expression in various kinds of human cancer, and its complex complicated regulatory networks affect the tumorigenicity and development of malignant tumors [34].

The seed uuggucc of miR-133b-3p contains three distinct nucleotides c, g and u. The associated card seq is the sequence [3,7,34,139,931,5208,⋯] that is away from that of the free group $F_{2}$. The Groebner basis attached to the seed contains the DP3 cubic surfaces $xyz+x^{2}+z^{2}-d=0$, d = 1 or 2, characteristic of the ADE type $A_{2}$, surfaces such as $xyz-yz+x^{2}+3=0$ of type $A_{2}A_{2}$ and surfaces of type $A_{1}$.

#### 2.1.13. The Oncomir miR-1270b

miR-1270 plays a crucial role in the initiation and development of osteosarcoma, which is the most common bone malignancy worldwide, characterized by high morbidity and mortality [35]. Incidentally, miR-1270 is also differentially expressed in Meniere’s disease, a chronic debilitating disorder of the inner ear, characterized by fluctuating episodes of vertigo and hearing loss [36].

The seed uggagau of miR-1270 contains three distinct nucleotides a, u and g. The associated card seq is that of the free group $F_{2}$. The Groebner basis attached to the seed contains DP3 surfaces such as $4xyz-4z^{2}+3y^{2}-2yz=0$ of ADE type $A_{4}$, $4xyz+4y^{2}+4z^{2}-z-2=0$ of type $A_{2}$ and $4xyz+3y^{2}-2yz-6=0$ of type $A_{2}A_{2}$. None of them belong to the Painlevé class

#### 2.1.14. The Oncomir miR-138-5p

miR-138-5p plays a critical role in the development of colorectal cancer [37].

The seed gcuggug of miR-138-5p contains three distinct nucleotides c, u and g. The associated card seq is that of the free group $F_{2}$. The Groebner basis attached to the seed contains DP3 surfaces of ADE type $A_{1}$ and $A_{2}$. One of them $xyz+3x^{2}+z^{2}+3=0$ is of type $A_{2}$ and in the Painlevé class $PV^{deg}$.

#### 2.1.15. The Oncomir miR-23a-5p

Dysregulation of miR-23a occurs in various human diseases, such as ischemia-reperfusion injury, coronary heart diseases and cancer [38].

The seed ggguucc of miR-23a-5p contains three distinct nucleotides c, u and g. The associated card seq is away from the free group $F_{2}$. The Groebner basis attached to the seed contains a DP3 surface $xyz+2x^{2}+2y^{2}+4=0$ which is of ADE type $A_{2}$ and in the Painlevé class $PV^{deg}$. Another surface $xyz+x^{2}+y^{2}+z^{2}-y=0$ is devoid of an isolated singularity but belongs to the Painlevé class.

#### 2.1.16. The Oncomir miR-328-3p

MiR-328-3p is dysregulated in various diseases, such as pulmonary arterial hypertension, multiple sclerosis and epilepsy, as well as cancers [39].

The seed uggcccu of miR-328-3p contains three distinct nucleotides c, u and g. The associated card seq is away from the free group $F_{2}$. The Groebner basis attached to the seed contains DP3 surfaces of the PVI type such as the Markoff surface $xyz+x^{2}+y^{2}+z^{2}=0$, the Fricke surface $xyz+x^{2}+y^{2}+z^{2}-3=0$ and other surfaces in the same class.

### 2.2. Human Micrornas That Mainly Act as Tumor Suppressors

Table 2 provides a short list of miRNAs generally considered as tumor suppressors. As already mentioned, the boundary between an oncomir and a tumor suppressor is not tight. The first five miRNAs either have a card seq away from the free group $F_{2}$ or have a Groebner basis containing simply singular surfaces in one of their -5-p or 3-p strands. However, many other miRNAs in Table 2 have the card seq of $F_{2}$ and no simply singular surface is found in the Groebner basis. This is in accordance of our approach.

#### 2.2.1. MicroRNA let-7a-5p

The let-7 family of microRNAs, first identified in C. elegans but functionally conserved from worms to human, is an important class of regulators for diverse cellular functions ranging from cell proliferation, differentiation and pluripotency to cancer development and progression [40,41]. let-7a is abundant in glioma and plays a role in prostate and lung cancer development. Lower expression of miR-let-7a in patients with lung cancer brain metastasis was closely related to unfavorable efficacy and prognosis of radiotherapy [42].

The seed gagguag of miR-let-7a-5p contains three distinct nucleotides a, g and u. The associated card seq is that of the free group $F_{2}$. The Groebner basis attached to the seed contains DP3 cubic surfaces $xyz+2z^{3}+3=0$ and $xyz+2z^{2}+xz+y-1=0$ of type $f^{\left(A_{1}A_{2}\right)}$ and $xyz+2z^{2}+xz+y+1=0$ of type $f^{\left(A_{1}A_{4}\right)}$. The later surface is pictured in Figure 4: left. None of these surfaces are found of the Painlevé type.

The seed uauacaa of miR-let-7a-5p contains three distinct nucleotides a, c and u. The associated card seq is that of the free group $F_{2}$. The Groebner basis attached to the seed does not contain cubics and higher order surfaces.

#### 2.2.2. MicroRNA 34b-3p

The overexpression of miR-34b-3p has been shown to suppress lung cancer cell growth, including proliferation inhibition, cell cycle arrest and increased apoptosis [43]. Downregulation of miR-34b-3p is significantly hypermethylated and consequently downregulated, particularly in neuroblastoma patients at high risk of progression [44].

The seed aaucacu of miR-34b-3p contains three distinct nucleotides a, c and u. The associated card seq is the sequence [3,10,51,164,1230,7829,⋯] that is away from that of the free group $F_{2}$. The Groebner basis attached to the seed contains DP3 surfaces $xyz+x^{2}+y^{2}-2=0$ which is of type $A_{2}$, as well as $xyz+x^{2}+y^{2}+yz-2y-1=0$ which is of type $A_{3}$ and $xyz+x^{2}+y^{2}+yz-1=0$ which is of type $A_{4}$. The later surface is pictured in Figure 4: right.

#### 2.2.3. MicroRNA 200a-5p

Expression in melanoma cells may be controlled by miR-200a-5p leading to a downregulation of surface expression for the human leukocyte antigen class I, which is linked to a reduced survival of patients [45]. The miR-200 family acts as an oncogene in colorectal cancer [46]. MicroRNAs from the miR-200 family are commonly associated with the inhibition of the metastatic potential of cancer cells, following inhibition of ZEB transcription factors expression and epithelial-to-mesenchymal transition [47].

The seed aucuuac of miR-200a-5p contains three distinct nucleotides a, c and u. The associated card seq is the sequence [3,10,35,140,921,5778,⋯] that is away from that of the free group $F_{2}$. The Groebner basis attached to the seed contains DP3 cubic surfaces of the ADE type $f^{\left(A_{1}A_{3}\right)}$ and $f^{\left(A_{1}A_{1}A_{3}\right)}$ which are not in the Painlevé class.

#### 2.2.4. MicroRNA 22-5p

A study revealed that miR-22-5p downregulation contributed to the malignant progression of non-small-cell lung cancer by targeting transcription factor TWIST2 [48]. miR-22-5p may also inhibit tumor progression by sponging DNA topoisomerase II-alpha [49].

The seed guucuuc of miR-22-5p contains three distinct nucleotides u, c and g. The associated card seq is that of the free group $F_{2}$. The Groebner basis attached to the seed contains a cubic surface which is the DP3 surface $y^{2}z+x^{2}-2y^{2}+2xy+1=0$ of the ADE type $D_{4}$.

#### 2.2.5. MicroRNA 214

miR-214 is one of the fastest evolving microRNAs ([50], Table 3). First, miR-214 was reported to promote apoptosis in HeLa cells. Presently, miR-214 is implicated in an extensive range of conditions such as cardiovascular diseases, cancers, bone formation and cell differentiation [51].

For miR-214-5p and the short seed sequence GCCUGU, we find the DP3 surface $xyz+y^{2}+z^{2}-4=0$ of type $A_{2}$. The reader can look at ([52], Figure 5) for a picture. A surface $xyz+y^{2}+z^{2}y-2z-1=0$ of $A_{2}$ is also found. For a longer seed GCCUGUC, surfaces are not found to contain a cubic surface.

For miR-214-3p and the short seed sequence cagcag, we find a DP3 surface $xyz+2xy-y^{2}+z+2=0$ of ADE type $f^{\left(A_{1}A_{4}\right)}$ in the corresponding Groebner basis. With the longer seed cagcagg, no cubic surface is found within the Groebner basis.

#### 2.2.6. MiRNA 503-5p

The slowest evolving miRNA gene in the human species is hsa-miR-503. It regulates gene expression in various pathological processes of diseases, including carcinogenesis, angiogenesis, tissue fibrosis, and colon cancer [53,54].

The seed region for miR-503-5p is agcagcg. The associated card seq is that of the free group $F_{2}$. The Groebner basis is a rational scroll devoid of isolated singularities. For a longer seed agcagcgg, singular surfaces are found ([52], Section 5).

#### 2.2.7. MicroRNA 141-5p

miR-141-5p, an important member of the miR-200 family, has been reported to be involved in cellular proliferation, migration, invasion, and drug resistance in different kinds of human malignant tumors. It may act as a tumor suppressor via targeting gene RAB32 in the development of chronic myeloid leukemia [55].

The seed aucuucc of miR-141-5p contains three distinct nucleotides a, u and c. The associated card seq is that of the free group $F_{2}$. The Groebner basis attached to the seed does not contain a cubic.

#### 2.2.8. MicroRNA 31-5p

miR-31 was shown to regulate a number of metastasis-related genes involved in proliferation, cell-cycle regulation and apoptosis [56]. miR-31-5p is a potential circulating biomarker and therapeutic target for oral cancer [57].

The seed ggcaaga of miR-31-5p contains three distinct nucleotides a, c and g. The associated card seq is that of the free group $F_{2}$. The Groebner basis attached to the seed contains a cubic surface $xy^{2}-2xy-x+z=0$ which is a rational scroll devoid of isolated singularity and not of the Painlevé type.

#### 2.2.9. MicroRNA 122-5p

miR-122 is used as a cutting-edge tool for cancer treatment. Cancer cell lines revealed that miR-122 functions as an oncogenic or tumor-suppressive miRNA and examination of patient tumor samples furthermore supported the idea that miR-122 dysregulation is a crucial component in the growth of malignancies [58].

The seed gagugug of miR-122-5p contains three distinct nucleotides a, u and g. The associated card seq is that of the free group $F_{2}$. The Groebner basis attached to the seed does not contain a cubic.

#### 2.2.10. MicroRNA 29b-5p

miR-29b-5p is a key oncomiR and therefore a target for molecular diagnosis and treatment of patients with oral squamous cell carcinoma [59].

The seed cugguuu of miR-29b-5p contains three distinct nucleotides c, u and g. The associated card seq is that of the free group $F_{2}$. The character variety attached to the seed contains cubics of a DP3 type, a rational scroll and a ruled surface over a genus 1 curve, but none of them has an isolated singularity, and they are not of the Painlevé type.

#### 2.2.11. MicroRNA 143-3p

Brain metastasis is one of the secondary mortality causes for patients. It was found that miR-143-3p is upregulated in the paired brain metastasis tissues in contrast with primary cancer tissues. It may be used as a prognostic factor for in vivo progression, invasion, metastasis and survival rate of lung cancer [60]. miR-143-3p may suppress tumorigenesis in pancreatic duct adenocarcinoma by targeting the KRAS gene [61].

The seed gagauga of miR-143-3p contains three distinct nucleotides a, u and g. The associated card seq is that of the free group $F_{2}$. The Groebner basis attached to the seed does not contain a cubic.

#### 2.2.12. MicroRNA 140-5p

The low expression of miR-140-5p is related to the tumor stage or metastasis. The miR-140-5p overexpression suppressed cell proliferation and invasion in colorectal carcinoma [62].

The seed agugguu of miR-140-5p contains three distinct nucleotides a, u and g. The associated card seq is that of the free group $F_{2}$. The Groebner basis attached to the seed contains a ruled surface over a genus 1 curve. There is no isolated singularity or a Painlevé type cubic.

#### 2.2.13. MicroRNA 340

miR-340 may act either as an oncogene or a tumor suppressor by targeting genes related to proliferation, apoptosis and metastasis, as well as those associated with diagnosis, treatment, chemoresistance and prognosis [63]. miR-340 restricts the development of breast cancer cells by targeting numerous oncogenes. miR-340 has been shown to play a suppressive role in lung, colorectal ovarian and prostate cancers, laryngeal squamous cell carcinoma and osteosarcoma. miR-340 plays a critical role in in non-small-cell lung cancer, and its overexpression restricts the growth and invasion of the corresponding cells [64].

The seed of miR-340-5p is uauaaag and that of miR-340-3p is ccgucuc. Both of them have a card seq which is that of the free group $F_{2}$. For both of them, the Groebner basis attached to the seed does not contain cubic surfaces.

## 3. Materials and Methods

For an introduction to Painlevé equations, readers are encouraged to explore our recent paper [10] and the references therein. Painlevé equations represent nonlinear ordinary differential equations whose solutions are considered transcendental due to their inability to be expressed in terms of familiar special functions, such as elliptic, hyperelliptic or hypergeometric functions [65]. The hallmark of Painlevé transcendents lies in the Painlevé property, indicating that the only movable singularities are poles [66]. In recent years, there has been a shift in focus towards uncovering explicit algebraic solutions of PVI, making it a significant subject in 20th-century mathematical research with connections to algebraic topology, algebraic geometry, and representation theory [67,68]. Painlevé equations of a lower index than VI exhibit a more complex topological structure, which is elaborated upon below. The topological structure and the associated $SL_{2}\left(C\right)$ representations of PVI, PV, and PIV are further elucidated. These features prove to be invaluable for understanding diseases at the transcriptome scale.

### 3.1. Painlevé Equations and Their Associated $SL_{2}\left(C\right)$ Character Variety

At a first step, the concept of a flat connection on a fiber bundle M→B is fully relevant, where the base *B* assumes the form of a three-punctured sphere, denoted as $B=S_{2}^{\left(3\right)}=P^{1}\setminus \left\{0,1,\infty \right\}$. Over a point t∈B, a corresponding four-punctured sphere $P_{t}=S_{2}^{\left(4\right)}=P^{1}\setminus \left\{0,1,t,\infty \right\}$ exists. Let $M_{t}$ denote the fiber of *M* over the base point t∈B, the monodromy action unfolds through the action of the fundamental group of the base on the fiber. This defines a homomorphism $\pi_{1}\left(B\right)\to Aut\left(M_{t}\right)$ [10,67].

For the setting of the monodromy problem for PVI, the reader may consult ([69], Section 1), where complex constants $\theta_{0}$, $\theta_{1}$, $\theta_{t}$ and $\theta_{\infty }$ arise from the eigenvalues of the matrices in the associated Schlesinger’s system.

For each t∈B, the space of conjugacy classes of $SL_{2}\left(C\right)$ representations for the fundamental group $\pi_{1}\left(P_{t}\right)≅F_{3}$ is the character variety
$$C_{t}=\mathrm{Hom}\left(\pi_{1}\left(P_{t}\right),G\right)/G,\;\;\mathrm{with}\;\;G=SL_{2}\left(C\right).$$

The connection is flat and described by $P_{VI}$ Equation as follows:(1)$$\begin{array}{cc} & y_{tt}=\frac{1}{2}\left(\frac{1}{y}+\frac{1}{y-1}+\frac{1}{y-t}\right)y_{t}^{2}-\left(\frac{1}{t}+\frac{1}{t-1}+\frac{1}{y-t}\right)y_{t}\\ & +\frac{y(y-1)(y-t)}{2t^{2}{\left(t-1\right)}^{2}}\left[\alpha +\beta \frac{t}{y^{2}}+\gamma \frac{t-1}{{\left(y-1\right)}^{2}}+\delta \frac{t(t-1)}{{\left(y-t\right)}^{2}}\right\}]\;\;\left(PVI\right) \end{array}$$
with $y_{t}=\frac{dy}{dt}$ and parameters $\alpha ={\left(\theta_{\infty }-1\right)}^{2}$, $\beta =-\theta_{0}^{2}$, $\gamma =\theta_{1}^{2}$, $\delta =1-\theta_{t}^{2}$.

Further topological steps are illustrated in Figure 1 ([19], Figure 3). The corresponding Painlevé equations are in [69] as follows:(2)$$y_{tt}=\left(\frac{1}{2y}+\frac{1}{y-1}\right)y_{t}^{2}-\frac{1}{t}y_{t}+\frac{{\left(y-1\right)}^{2}}{t^{2}}\left(\alpha y+\frac{\beta }{y}\right)+\frac{\gamma y}{t}+\frac{\delta y(y+1)}{y-1}\;\;\left(PV\right)$$
with parameters $\alpha =\frac{1}{8}{\left(\theta_{0}-\theta_{1}+\theta_{\infty }\right)}^{2}$, $\beta =-\frac{1}{8}{\left(\theta_{0}-\theta_{1}-\theta_{\infty }\right)}^{2}$, $\gamma =1-\theta_{0}-\theta_{1}$, $\delta =-\frac{1}{2}$.
(3)$$y_{tt}=\frac{1}{2y}y_{t}^{2}+\frac{3}{2}y^{3}+4ty^{2}+2\left(t^{2}-\alpha \right)y+\frac{\beta }{y}\;\;\left(PIV\right)$$
with parameters $\alpha =2\theta_{\infty }-1$, $\beta =-8\theta_{0}^{2}$.
(4)$$y_{tt}=2y^{3}+ty+\alpha \;\;\left(PII\right)$$
with parameter $\alpha =\frac{1}{2}-\theta_{0}$.

#### The Painlevé–Jimbo–Fricke Cubics Attached to Painlevé Equations

In the following, we introduce the boundary traces *a*, *b*, *c*, *d* of matrices representing simple loops around the punctures of the corresponding Riemann surface shown in Figure 1. In papers [19,70], it is shown that the $SL_{2}\left(C\right)$ character variety takes the form of Painlevé–Jimbo–Fricke cubics as follows. It should be mentioned that, from now, the space variables *x*, *y*, *z* are used for defining surfaces, not the Painlevé equations.
(5)$$\begin{array}{cc} & xyz+x^{2}+y^{2}+z^{2}-\omega_{1}x-\omega_{2}y-\omega_{3}z-\omega_{4}=0\;\;,\\ & \omega_{1}=ad+bc,\;\omega_{2}=ac+bd,\;\omega_{3}=ab+cd,\;\omega_{4}=a^{2}+b^{2}+c^{2}+d^{2}+abcd-4,\\ & \left(type\;PVI\right)\\ & xyz+x^{2}+y^{2}-\omega_{1}x-\omega_{2}y-\omega_{3}z-\omega_{4}=0,\;\;\\ & \;\;\mathrm{with}\;\;\omega_{4}=-1-\omega_{3}^{2}-\frac{\omega_{3}\left(-\omega_{2}+\omega_{1}\omega_{3}\right)\left(-\omega_{1}+\omega_{2}\omega_{3}\right)}{{\left(\omega_{3}^{2}-1\right)}^{2}},\\ & \omega_{1}=a+bc,\;\omega_{2}=b+ac,\;\omega_{3}=c,\;\omega_{4}=-\left(abc+c^{2}+1\right)\\ & \left(type\;PV\right)\\ & xyz+x^{2}+y^{2}-\omega_{1}x-\omega_{2}y+1=0\;\;,\\ & \omega_{1}=a+bc,\;\omega_{2}=b,\\ & \left(type\;PV_{deg}\right)\\ & xyz+x^{2}-\omega_{1}x-\omega_{2}\left(y+z\right)-\omega_{2}\left(-\omega_{1}+\omega_{2}\right)=0,\\ & \omega_{1}=ab+b^{2},\;\omega_{2}=b^{2},\\ & (type\;PIV)\;\;\\ & xyz+x^{2}-\omega_{1}x-y+1=0\;\;.\\ & \omega_{1}=a,\\ & (type\;PII^{FN}). \end{array}$$
The Fricke cubics refers to the case of PVI character variety [71]. Equations in array (Equation 5) correspond to Painlevé Equations (Equation 1)–(Equation 4).

### 3.2. The Character Variety Associated to a DNA/RNA Sequence

We can consider the representations of an $f_{p}$ group over the space-time-spin group $SL_{2}\left(C\right)$, as we did in [6,7].

Representations of $f_{p}$ in $SL_{2}\left(C\right)$ are homomorphisms $\rho :f_{p}\to SL_{2}\left(C\right)$ with character $\kappa_{\rho }\left(g\right)=tr\left(\rho \left(g\right)\right)$, $g\in f_{p}$. The notation tr(ρ(g)) signifies the trace of the matrix ρ(g). The set of characters is employed to determine an algebraic set by taking the quotient of the set of representations ρ by the group $SL_{2}\left(C\right)$, which acts by conjugation on representations [71,72].

In our paper, ([6], Section 2.2), we elaborated that the character variety of $f_{p}$ is a set comprised a sequence *X* of multivariate polynomials. A particular basis related to *X* is the Groebner basis G(X), whose factors define hypersurfaces.

For a two-generator group $f_{p}$, the factors are three-dimensional surfaces. In general, these surfaces can be classified by mapping them to a rational surface across five categories ([6], Section 3). Often encountered surfaces are degree *p* Del Pezzo surfaces where 1≤p≤9. A rational surface may either be non-singular, ‘almost non-singular’, having only isolated singularities, or singular. Almost non-singular surfaces are crucial in our context. A simple singularity is referred to as an A-D-E singularity and must be of the type $A_{n}$, n≥1, $D_{n}$, n≥4, $E_{6}$, $E_{7}$ or $E_{8}$.

The A-D-E type is mirrored in the notation we employ. For instance, $S^{\left(lA_{1},mA_{2},nA_{3},\cdots \right)}$ denotes a surface containing *l* type $A_{1}$, *m* type $A_{2}$, *n* type $A_{3}$ singularities, etc. A generic surface is the Cayley cubic we encountered in our previous papers, defined as $S^{\left(4A_{1}\right)}=xyz+x^{2}+y^{2}+z^{2}-4$ ([6], Figure 5).

For a three-generator group $f_{p}$, the factors of G(X) are seven-dimensional surfaces of the form $S_{a,b,c,d}\left(x,y,z\right)$. Some of them belong to the Fricke family ([6], Equation (3)), which is associated with the four-punctured sphere. However, for a chosen set of parameters a,b,c,d, the hypersurface reduces to an ordinary three-dimensional surface.

For a four-generator group $f_{p}$, the factors of G(X) are 14-dimensional surfaces generally containing 4 copies of the form S(x,y,z), S(x,u,v), S(y,u,v) and S(z,v,w) for selected choices of 8 parameters.

For the calculations of the character variety, we make use of a software on Sage [73]. We also need Magma [74] in the calculation of a Groebner basis for 3-base sequences.

## 4. Conclusions

In this paper, we proposed a group theoretical definition of homeostasis at the level of the transcriptome. We applied this research to miRNAs controlling the (dys)regulation of gene expression in cancer. In our approach, the seed of a miRNA serves as a generator for an infinite group $f_{p}$ whose $SL_{2}\left(C\right)$ character variety, more precisely its Groebner basis G, contains cubic surfaces that are a signature of a possible disease. We found a correlation between oncomirs and either (i) $f_{p}$ is away from the free group $F_{r}$, or (ii) G contains at least a simply singular cubic $f^{\left(ADE\right)}$, or (iii) the dynamics of $f^{\left(ADE\right)}$ is a non-linear differential equation of the Painlevé type.

We emphasized the connection of the Painlevé VI equation to the topology of the four-punctured sphere and its breaking when two punctures collapse. Mathematically, the character variety of PVI is described by a fundamental group. In contrast, the character variety of the PV equation is a fundamental groupoid that rules the ‘Stokes phenomenon’ of breaking the four-punctured sphere [13,75].

The model is applied to many mirRNAs known to play a role in cancer. In the future, this approach may be applied to other nucleotide sequences that play a role in the regulation of gene expression. The next goal of this research is to investigate the gene targets of miRNAs [76]. A miRNA-target network may be described from a statistical tool such as MiEnTurNet [77]. The output is a bipartite graph with its vertices colored alternately black and white. The graph may be embedded in an oriented (Riemann) surface having the structure of an algebraic curve over the field of algebraic numbers. This is called a dessin d’enfant (child’s drawing), a concept developed by Alexandre Grothendieck [78].

Black points are seeds of the miRNAs, and white points are the target genes. A permutation group describes the symmetries of the graph.

## Figures and Tables

**Figure 1 ijms-25-04971-f001:**
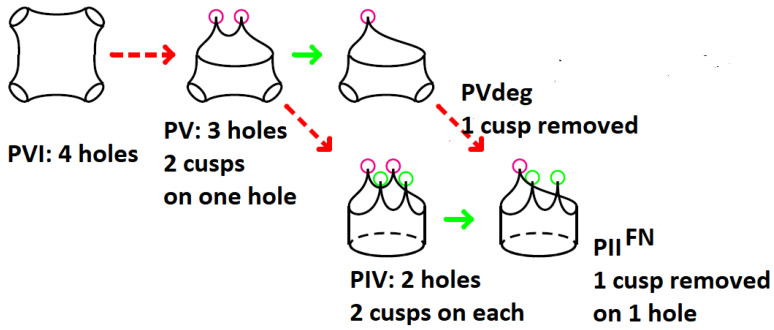
A partial view of singularities of Riemann surfaces from the Painlevé perspective ([19], Figure 3). The dashed arrows correspond to the creation of cusps (red or green bullets) on some holes (black bullets). The full arrows point out a cusp removal. Only Painlevé topologies expected in the context of our model of gene expression are considered. The paths in the diagram correspond to the move PVI → PV → PIV → ${\mathrm{PII}}^{FN}$ or the move PVI → PV → ${\mathrm{PV}}_{deg}$→ ${\mathrm{PII}}^{FN}$.

**Figure 2 ijms-25-04971-f002:**
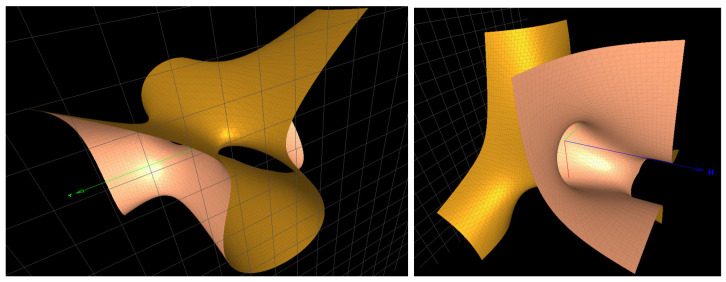
(**Left**): the surface $x^{3}-y^{2}z+xy-3x+z=0$ of type $f^{\left(A_{2}\right)}$ in the Groebner basis for the character variety of $G_{miR-21-3p}$. (**Right**): the surface $xyz-x^{2}-y^{2}-yz+x+2=0$ of the ADE type $f^{\left(A_{2}\right)}$ in the Groebner basis for $G_{miR-204-5p}$.

**Figure 3 ijms-25-04971-f003:**
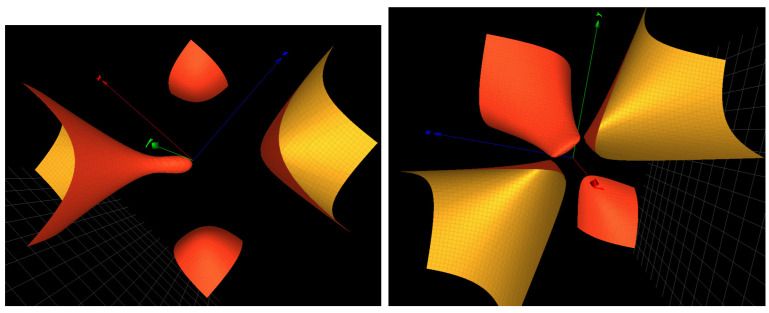
(**Left**): the surface $xyz+x^{2}+y^{2}+z^{2}-2y+z+1=0$ of type PVI in the Groebner basis for the seed of miR-9-5p. (**Right**): the surface $xyz+y^{2}+z^{2}-2y-2z+1=0$ of type $f^{\left(A_{2}A_{1}A_{1}\right)}:PV$ in the Groebner basis for the seed of miR-146b-3p.

**Figure 4 ijms-25-04971-f004:**
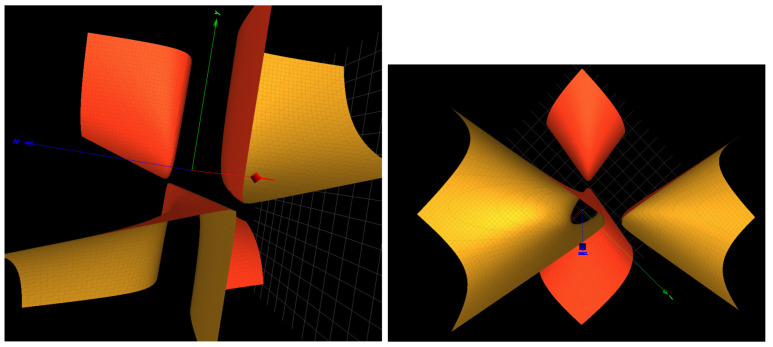
(**Left**): the surface $xyz+2z^{2}+xz+y+1=0$ of type $f^{\left(A_{1}A_{4}\right)}$ in the Groebner basis for the seed of miR-let-7a. (**Right**): the surface $xyz+x^{2}+y^{2}+yz-1=0=0$ of type $f^{\left(A_{4}\right)}$ in the Groebner basis for the seed of miR-34b-3p.

**Table 1 ijms-25-04971-t001:** A partial list of miRNAs generally considered as oncomirs.

Oncomir	Seed	Card Seq	Sing Type	Cancer Type
miR-21-3p	aacacca	not $F_{1}$	$f^{\left(A_{3}\right)}$, $f^{\left(A_{2}\right)}$	ubiqitous
miR-204-5p	ucccuuu	not $F_{1}$	$f^{\left(A_{3}\right)}$, $f^{\left(A_{2}\right)}$	prostate
miR-126-5p	auuauua	$H_{3}$	$f^{\left(A_{2}\right)}$	liver
miR-1908-5p	ggcgggg	$F_{1}$	$f^{\left(4A_{1}\right)}$: PVI	ubiquitous
miR-1908-3p	cggccgc	$H_{3}$	$f^{\left(A_{2}\right)}$, $f^{\left(A_{1}A_{2}\right)}$	breast
miR-155-3p	uccuaca	not $F_{2}$	$f^{\left(A_{2}\right)}$, $f^{\left(A_{1}A_{2}\right)}$	ubiquitous
miR-9-5p	cuuuggu	$F_{2}$	$f^{\left(A_{2}\right)}$: ${\mathrm{PV}}^{deg}$, PVI	metastasis
miR-146b-3p	cccugug	not $F_{2}$	$f^{\left(A_{2}\right)},f^{\left(A_{1}A_{1}A_{2}\right)}$}: PV	pancreatic
miR-19a-5p	guuuugc	$F_{2}$	$f^{\left(A_{2}A_{2}\right)}$, $f^{\left(A_{1}A_{1}\right)}$, ${\mathrm{PVI}}_{a}$	pancreatic
miR-181a-5p	acauuca	$F_{2}$	$f^{\left(A_{2}\right)}$,$f^{\left(A_{4}\right)}$	ubiquitous
miR-15-5p	agcagca	$F_{2}$	$\left\{f^{\left(A_{2}\right)},f^{\left(A_{1}A_{2}\right)}\right\}$: PV	prostate
miR-569	guuaaug	not $F_{2}$	$f^{\left(A_{2}\right)},f^{\left(A_{4}\right)}$	lung, breast
miR-133b	uuggucc	not $F_{2}$	$f^{\left(A_{2}\right)}$, $f^{\left(A_{1}\right)}$, $f^{\left(A_{2}A_{2}\right)}$	ubiquitous
miR-1270	uggagau	$F_{2}$	$f^{\left(A_{2}\right)}$, $f^{\left(A_{2}A_{2}\right)}$	osteosarcoma
miR-138-5p	gcuggug	$F_{2}$	$f^{\left(A_{1}\right)}$, $f^{\left(A_{2}\right)}$, $f^{\left(A_{2}\right)}$: ${\mathrm{PV}}^{deg}$	colorectal
miR-23a-5p	ggguucc	not $F_{2}$	$f^{\left(A_{2}\right)}$: ${\mathrm{PV}}^{deg}$, PVI	ubiquitous
miR-328-3p	uggcccu	not $F_{2}$	PVI	ubiquitous

**Table 2 ijms-25-04971-t002:** A partial list of miRNAs generally considered as tumor suppressors. According to our view, as well as the references given in the text, the first five miRNAs behave more likely as oncomirs.

Tumor Suppressor	Seed	Card Seq	Sing Type	Cancer Type
miR-let-7a-5p	gagguag	$F_{2}$	$f^{\left(A_{2}A_{2}\right)}$, $f^{\left(A_{1}A_{4}\right)}$	ubiquitous
miR-let-7a-3p	uauacaa	$F_{2}$	no cubic	ubiquitous
miR-34b-3p	aaucacu	not $F_{2}$	$f^{\left(A_{2}\right)}$, $f^{\left(A_{3}\right)}$, $f^{\left(A_{4}\right)}$	lung, breast
miR-200a-5p	aucuuac	not $F_{2}$	$f^{\left(A_{1}A_{3}\right)}$, $f^{\left(A_{1}A_{1}A_{3}\right)}$	skin, colorectal
miR-22-5p	guucuuc	$F_{2}$	$f^{\left(D_{4}\right)}$	nsc lung
miR-214-5p	gccugu	$F_{2}$	$f^{\left(A_{2}\right)}$	ubiquitous
.	gccuguc	$F_{2}$	no cubic	.
miR-214-3p	cagcag	$F_{2}$	$f^{\left(A_{1}A_{4}\right)}$	.
.	cagcagg	$F_{2}$	no cubic	.
miR-503-5p	agcagcg	$F_{2}$	rational scroll	colon
miR-141-5p	aucuucc	$F_{2}$	no cubic	blood leukemia
miR-31-5p	ggcaaga	$F_{2}$	rational scroll	oral
miR-122-5p	gagugug	$F_{2}$	no cubic	ubiquitous
miR-29b-5p	cugguuu	$F_{2}$	no isol. sing.	head, neck
miR-143-3p	gagauga	$F_{2}$	no cubic	pancreatic
miR-140-5p	agugguu	$F_{2}$	no isol. sing.	colorectal
miR-340-5p	uauaaag	$F_{2}$	no cubic	ubiquitous
miR-340-3p	ccgucuc	$F_{2}$	no cubic	ubiquitous

## Data Availability

Data are available from the authors upon reasonable request.

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
