# Peer review of "Topology and Dynamics of Transcriptome (Dys)Regulation"

_ijms, 2024, doi:10.3390/ijms25094971_

Round 1

Reviewer 1 Report

Comments and Suggestions for Authors

In the scope of IJMS journal I read “Fundamental theoretical problems of broad interest in biology, chemistry and medicine ” and “Application of the theories and novel technologies to specific experimental studies and calculations”. The paper under examination fits well these two items.

The general idea is that a short DNA or RNA sequence rel may be ‘healthy’ or disruptive (potentially leading to a disease) in a relevant context.  Taking the sequence as the generator of an infinite group fp on 2 to 4 letters/nucleotides A, U/T ,G and C, a healthy sequence should obey some ‘homeostasis’ rules (a) fp should close to a free group, (b) the sequence should be aperiodic, (c) the SL2(C) character variety attached to fp, more precisely its Groebner basis G should not contain simply singular cubic surfaces.

At a further step of the theory, the authors observe that some of such surfaces are governed by Painlev´e equations that are nonlinear transcendental ordinary differential equations.

Section 2 is an up to date account of the topology underlying Painlev´e equations of type PVI, PV and PIV that are relevant in the aforementioned context. In Section 3, the authors concentrate on the part of the post-transcriptome regulated by microRNA’s appearing in cancer research.

The manuscript is intriguing, in particular concerning the issue of theoretical biology of gene expression and regulation.  

I recommend the publication of the paper with minor revisions after the authors account for the following remarks:

1)       What are the next steps of this research? The authors may indicate them in the conclusion.

2)       The paper ‘About miR-NAs, miRNA seeds, target genes and target pathways ’, by Tim Kehl et al, Oncotarget, 2017, Vol. 8, (No. 63), pp: 107167-107175 and the relatedon-line facility for identifying the genee targets and pathways for miRNAs  must be quoted and widely discussed.

3)       In “Materials and Methods”, the Authors stated that “Painlevé equations represent nonlinear ordinary differential equations whose solutions are considered transcendental due to their inability to be expressed in terms of familiar special functions, such as elliptic, hyperelliptic, or hypergeometric functions”. However, the elliptic functions are defined by second-order ordinary differential equations whose singularities have the Painlevé property: the only movable singularities are poles.  Please explain this seeming contradiction.

4)       Figure 1 is rather rough and need to be fully redrawn.

5)       The discussion must be separated from the results, otherwise the presentation is confusing.

6)       Cross the boundaries of disciplines is always a difficult task.  The question remains about the intended audience: either mathematicians interested in biology, or biologists interested in getting new tools, or both.

Comments on the Quality of English Language

The English is good enough. 

Author Response

Thank you for the useful comments and recommendations.

1) In the conclusion, we added the sentence:
The next goal of this research is to investigate the gene targets of miRNAs. A miRNA-target network may be described from a statistical tool such as
MiEnTurNet: http://userver.bio.uniroma1.it/apps/mienturnet/. The output is a bipartite graph with its vertices colored alternately black and white.
The graph may be embedded in an oriented (Riemann) surface having the structure of an algebraic curve over the field of algebraic numbers. The graph
is called a dessin d’enfant (child’s drawing), a concept developed by Alexandre Grothendieck: https://en.wikipedia.org/wiki/Dessin d’enfant. Black
points are seeds of the miRNAs and white points are the target genes. A permutation group describes the symmetries of the graph.

2)
We added the reference to the paper  ‘About miRNAs, miRNA seeds, target genes and target pathways ’, by Tim Kehl et al, Oncotarget, 2017, Vol. 8, (No. 63), pp: 107167-107175. 
This will be discussed at length in our next paper: see item 1.

3)
There is no contradiction. Most solutions are not of the elliptic type.

4)
Figure 1 was redrawn.

5)
We did not follow this recommendation because our discussion is about the obtained results about the miRNAs listed in Tables 1 and 2. 
In our opinion, the presentation is natural as is.

6)
We hope that both audiences are concerned.

Reviewer 2 Report

Comments and Suggestions for Authors

The study provides contributions in the identification of disruptive short sequences in RNA transcription and regulation to treat disease. The examples in the study show the equations and mathematical structures within the context of miRNAs.

The following suggestions can be recommended:

The dataset attributes can be provided in more detail.

Proofreading can be done. Ribonucleic acid (RNA) can be first written in its full form in the part it first appears in the text.

The contractions like “We’ve previously….” Can be eliminated. “We have….” can be written instead of the contractions.

Editing can also be done to make some terms more aligned with academic terminology.

Limitations, if applicable, can be stated.

The last sentence in the abstract can be explained in more detail and / or specificity: “These findings should find applications in cancer research.      

The distinctive and novel parts of the paper can be indicated by comparing the work and its scheme with other works in the literature.

If software has been used, it should be indicated. It can be also shown in the references.

Finally, equations can be checked to make sure all the equations have been cited in the text.

Yours faithfully,

Comments on the Quality of English Language

Some changes required have been provided in the comments to the authors part. 

Author Response

Thank you for the useful comments and recommendations.

"The dataset attributes can be provided in more detail."
The data set comes from the Mir base in Reference [29] which is a very standard repository. We write it in a more explicit way 
at the beginning of Section 3.

""Proofreading can be done. Ribonucleic acid (RNA) can be first written in its full form in the part it first appears in the text."
We did it.

"The contractions like “We’ve previously….” Can be eliminated. “We have….” can be written instead of the contractions."
You are right.

"Editing can also be done to make some terms more aligned with academic terminology."
We failed to do better, may be because we are not experts in the community of molecular biology.

"Limitations, if applicable, can be stated."
Of course there are limitations of the theory that can be better investigated in the future.

"The last sentence in the abstract can be explained in more detail and / or specificity: “These findings should find applications in cancer research. "
Done     

"The distinctive and novel parts of the paper can be indicated by comparing the work and its scheme with other works in the literature."
The present work and the related references to our own papers are truly new. Hopefully our research will be followed in the future by other groups.

"If software has been used, it should be indicated. It can be also shown in the references."
Yes, this was added as references [23] and [24].

"Finally, equations can be checked to make sure all the equations have been cited in the text."
Yes, we did it.